# Competition over data: how does data purchase affect users?

**Yongchan Kwon**  *yk3012@columbia.edu*
*Department of Statistics, Columbia University*

**Tony Ginart**  *tginart@stanford.edu*
*Department of Electrical Engineering, Stanford University*

**James Zou**  *jamesz@stanford.edu*
*Department of Biomedical Data Science, Stanford University*

**Reviewed on OpenReview:** *https: // openreview. net/ forum? id= 63sJsCmq6Q*

## Abstract

As the competition among machine learning (ML) predictors is widespread in practice, it becomes increasingly important to understand the impact and biases arising from such competition. One critical aspect of ML competition is that ML predictors are constantly updated by acquiring additional data during the competition. Although this active data acquisition can largely affect the overall competition environment, it has not been well-studied before. In this paper, we study what happens when ML predictors can purchase additional data during the competition. We introduce a new environment in which ML predictors use active learning algorithms to effectively acquire labeled data within their budgets while competing against each other. We empirically show that the overall performance of an ML predictor improves when predictors can purchase additional labeled data. Surprisingly, however, the quality that users experience—*i.e.*, the accuracy of the predictor selected by each user—can decrease even as the individual predictors get better. We demonstrate that this phenomenon naturally arises due to a trade-off whereby competition pushes each predictor to specialize in a subset of the population while data purchase has the effect of making predictors more uniform. With comprehensive experiments, we show that our findings are robust against different modeling assumptions.

## 1 Introduction

When there are several companies on a marketplace offering similar services, a customer usually chooses the one that offers the best options or functionalities. It naturally causes competition among the companies, and they are motivated to offer high-quality services, as their ultimate goal is to attract more customers and make more profits. When it comes to machine learning (ML)-based services, high-quality services are often achieved by re-training their models after acquiring more data from customers or data vendors (Meierhofer et al., 2019). In this paper, we consider a competition situation where multiple companies offer ML-based services while constantly improving their predictions by acquiring labeled data.

For instance, we consider the U.S. auto insurance market (Jin & Vasserman, 2019; Sennaar, 2019). The auto insurance companies including State Farm, Progressive, and AllState use ML models to analyze customer data, assess risk, and adjust actual premiums. These companies also offer insurance called the Pay-How-You-Drive, which is usually cheaper than regular auto insurances on the condition that the insurer monitors driving patterns, such as rapid acceleration or oscillations in speed (Arumugam & Bhargavi, 2019; Jin & Vasserman, 2019). That is, the companies essentially provide financial benefits to customers, collecting customers' driving pattern data. With these user data, they can regularly update their ML models, improving model performance while competing with each other.

**Case 1. There is a buyer**

**Case 2. No one shows purchase intent**

Figure 1: Illustrations of our competition environment (left) when there is a company showing purchase intent and (right) when no company shows purchase intent. In step 1, described in the first arrow, each predictor receives a user query and decides whether to buy user data. In step 2, described in the second arrow, (left) if there is a company that thinks the data is worth buying, the company shows purchase intent. The user $X_t$ then selects the buyer for financial benefits. (Right) If no one thinks the user data is worth buying, a user selects one company based on received ML predictions. In step 3, the only selected predictor gets the user label $Y_t$ and updates its model. We provide details on the environment in Section 2.

Analyzing the effects of data purchase in competitions could have practical implications, but it has not been studied much in the ML literature. The effects of data acquisition have been investigated in active learning (AL) literature (Settles, 2009; Ren et al., 2020), but it is not straightforward to establish competition in AL settings because it considers the single-agent situation. Recently, Ginart et al. (2021) studied implications of competitions by modeling an environment where ML predictors compete against each other for user data. They showed that competition pushes competing predictors to focus on a small subset of the population and helps users find high-quality predictions. Although this work describes an interesting phenomenon, it is limited to describe the data purchase system due to the simplicity of its model. The impact of data purchases on competition has not been studied much in the literature, which is the main focus of our work. Our environment is able to model situations where competing companies actively acquire user data by providing a financial benefit to users, and influence the way users choose service providers (See Figure 1). Related works are further discussed in Section 1.1.

**Contributions**  In this paper, we propose a general competition environment and study what happens when competing ML predictors can actively acquire user data. Our main contributions are as follows.

- We propose a novel environment that can simulate various real-world competitions. Our environment allows ML predictors to use AL algorithms to purchase labeled data within a finite budget while competing against each other (Section 2).

- Surprisingly, our results show that when competing ML predictors purchase data, the quality of the predictions selected by each user can decrease even as competing ML predictors get better (Section 3.1).

- We demonstrate that data purchase makes competing predictors similar to each other, leading to this counterintuitive finding (Section 3.2). Our finding is robust and is consistently observed across various competition situations (Section 3.3).

- We theoretically analyze how the diversity of a user's available options can affect the user experience to support our empirical findings. (Section 4).

## 1.1 Related works

This work builds off and extends the recent paper, Ginart et al. (2021), which studied the impacts of the competition. We extend this setting by incorporating the data purchase system into competition systems.

Note that the setting by Ginart et al. (2021) is a special case of ours when no competitor buys user data, *i.e.*, $n_{\mathrm{b}}^{(i)} = 0$ for all $i \in [M]$ with the notation in Section 2. Our environment enables us to study the impacts of data acquisition in competition, which is not considered in the previous work. Compared to the previous work, which showed competing predictors become too focused on sub-populations, our work suggests that this can be a good thing in the sense that it provides a variety of different options and better quality of the predictors selected by users.

A related field of our work is the stream-based AL, the problem of learning an algorithm that effectively finds data points to label from a stream of data points. (Settles, 2009; Ren et al., 2020). AL has been shown to have better sample complexity than passive learning algorithms (Kanamori, 2007; Hanneke et al., 2011; El-Yaniv & Wiener, 2012), and it is practically effective when the training sample size is small (Konyushkova et al., 2017; Gal et al., 2017; Sener & Savarese, 2018). However, our competition environment is significantly different from AL. In AL, since there is only one agent, competition cannot be established. In addition, while an agent in AL collects data only from label queries, competing predictors in our environment can obtain data from data purchase as well as regular competition. These differences create a unique competition environment, and this work studies the impacts of data purchase in competitive systems.

Competition has been studied in multi-agent reinforcement learning (MARL), which is a problem of optimizing goals in a setting where a group of agents in a common environment interact with each other and with the environment (Lowe et al., 2017; Foerster et al., 2017). Competing agents in MARL maximize their own objective goals that could conflict with others. This setting is often characterized by zero-sum Markov games and is applied to video games such as Pong or Starcraft II (Littman, 1994; Tampuu et al., 2017; Vinyals et al., 2019). As for the theoretical works, on zero-sum Markov games, Bai & Jin (2020) studies a sublinear regret algorithm, and Loftin et al. (2021) studies the efficient exploration in competitive multi-agent settings. We refer to Busoniu et al. (2008) and Zhang et al. (2019) for a complementary literature survey of MARL.

Although MARL and our environment have some similarities, the user selection and the data purchase in our environment uniquely define the interactions between users and ML predictors. In MARL, it is assumed that all agents observe information drawn from the shared environment. Different agents may observe different statuses and rewards, but all agents receive information and use them to update the policy function. In contrast, in our environment, the only selected predictor obtains the label and updates the predictor, which might be more realistic. In addition, ML predictors can collect data points through the data purchase. These settings have not been considered in the field of MARL.

## 2   A general environment for competition and data purchase

This section formally introduces a new and general competition environment. In our environment, competition is represented by a series of interactions between a sequence of users and fixed competing ML predictors. Here the interaction is modeled by supervised learning tasks.

**Notations**   At each round $t \in [T] := \{1, \ldots, T\}$, we denote a user query by $X_t \in \mathcal{X}$ and its associated user label by $Y_t \in \mathcal{Y}$. We focus on classification problems, *i.e.*, $|\mathcal{Y}|$ is finite, while our environment can easily extend to regression settings. We denote a sequence of users by $\{(X_t, Y_t)\}_{t=1}^{T}$ and assume users are independent and identically distributed (i.i.d.) by some distribution $P_{X,Y}$. We call $P_{X,Y}$ the user distribution.

As for the ML predictor side, we suppose there are $M$ competing predictors in a market. For $i \in [M]$, each ML predictor is described as a tuple $\mathcal{C}^{(i)} := (n_{\mathrm{s}}^{(i)}, n_{\mathrm{b}}^{(i)}, f^{(i)}, \pi^{(i)})$, where $n_{\mathrm{s}}^{(i)} \in \mathbb{N}$ is the number of i.i.d. seed data points from $P_{X,Y}$, $n_{\mathrm{b}}^{(i)} \in \mathbb{N}$ is a budget, $f^{(i)} : \mathcal{X} \to \mathcal{Y}$ is an ML model, and $\pi^{(i)} : \mathcal{X} \to \{0, 1\}$ is a buying strategy. We consider the following setting. A predictor $\mathcal{C}^{(i)}$ initially owns $n_{\mathrm{s}}^{(i)}$ data points and can additionally purchase user data within $n_{\mathrm{b}}^{(i)}$ budgets. We set the price of one data point is one, *i.e.*, a predictor $\mathcal{C}^{(i)}$ can purchase up to $n_{\mathrm{b}}^{(i)}$ data points from a sequence of users. A predictor $\mathcal{C}^{(i)}$ produces a prediction using the ML model $f^{(i)}$ and determines whether to buy the user data with the buying strategy $\pi^{(i)}$. We consider the utility function for $\mathcal{C}^{(i)}$ is the classification accuracy of $f^{(i)}$ with respect to the user distribution $P_{X,Y}$. Lastly, $f^{(i)}$ and $\pi^{(i)}$ are allowed to be updated throughout the $T$ competition rounds. That is, companies keep improving their ML models with newly collected data points.

**Competition dynamics**    Before the first competition round, all the $M$ competing predictors independently train their model $f^{(i)}$ using the $n_s^{(i)}$ seed data points. After this initialization, at each round $t \in [T]$, a user sends a query $X_t$ to all the predictors $\{\mathcal{C}^{(j)}\}_{j=1}^M$, and each predictor $\mathcal{C}^{(i)}$ determines whether to buy the user data. We describe this decision by using the buying strategy $\pi^{(i)}$. If the predictor $\mathcal{C}^{(i)}$ thinks that the labeled data would be worth one unit of budget, we denote this by $\pi^{(i)}(X_t) = 1$. Otherwise, if $\mathcal{C}^{(i)}$ thinks that it is not worth one unit of budget, then $\pi^{(i)}(X_t) = 0$. As for the $\pi^{(i)}$, ML predictors can use any stream-based AL algorithm (Freund et al., 1997; Žliobaitė et al., 2013). For instance, a predictor $\mathcal{C}^{(i)}$ can use the uncertainty-based AL rule (Settles & Craven, 2008)—*i.e.*, $\mathcal{C}^{(i)}$ attempts to purchase user data if the current prediction $f^{(i)}(X_t)$ is not confident (e.g., the Shannon's entropy of $p^{(i)}(X_t)$ is higher than some predefined threshold value where $p^{(i)}(X_t)$ is the probability estimate at the $t$-th round). In brief, we suppose a predictor $\mathcal{C}^{(i)}$ shows purchase intent if the remaining budget is greater than zero and $\pi^{(i)}(X_t) = 1$. If the remaining budget is zero or $\pi^{(i)}(X_t) = 0$, then $\mathcal{C}^{(i)}$ does not show purchase intent and suggests a prediction $f^{(i)}(X_t)$ to the user.

We now elaborate on how a user selects one predictor. At every round $t \in [T]$, the user selects only one predictor based on both purchase intents and prediction information received from $\{\mathcal{C}^{(j)}\}_{j=1}^M$. If there is a buyer, then we assume that a user selects one of the companies with purchase intent and receives the prediction he or she selects. We can think of this as a bargain in that the company offers a financial advantage (e.g., discounts or coupons) and the user selects it even if the quality might not be the best. When there is more than one buyer, we assume a user selects one of them uniformly at random. Once selected, the only selected predictor's budget is reduced by one; all other predictor's budget stays the same because they are not selected and do not have to provide financial benefits. If no predictor shows purchase intent, then a user receives prediction information $\{f^{(j)}(X_t)\}_{j=1}^M$ and chooses the predictor $\mathcal{C}^{(i)}$ with the following probability.

$$P\left(W_t = i \mid Y_t, \{f^{(j)}(X_t)\}_{j=1}^M\right) = \frac{\exp\left(\alpha q\left(Y_t, f^{(i)}(X_t)\right)\right)}{\sum_{j=1}^M \exp\left(\alpha q\left(Y_t, f^{(j)}(X_t)\right)\right)}, \tag{1}$$

where $\alpha \geq 0$ denotes a temperature parameter and $W_t \in [M]$ denotes the index of selected predictor. Here, $q : \mathcal{Y} \times \mathcal{Y} \to \mathbb{R}^+ := \{z \in \mathbb{R} \mid z \geq 0\}$ is a predefined quality function that measures similarity between the user label $Y_t$ and the prediction (e.g., $\mathbb{1}(\{Y_1 = Y_2\})$). With the softmax function in Equation (1), users are more likely to select high-quality predictions, describing the *rationality* of the user selection. Here, the temperature parameter $\alpha$ indicates how selective users are. For instance, $\alpha$ is close to $\infty$, users are very confident in their selection and choose the best company. Afterwards, the selected predictor $\mathcal{C}^{(W_t)}$ gets the user label $Y_t$ and updates the model $f^{(W_t)}$ by training on the new datum $(X_t, Y_t)$. The other predictors $f^{(i)}$ stay the same for $i \neq W_t$. We describe our competition system in Environment 1.

**Characteristics of our environment**    Our environment simplifies real-world competition and data purchases, which usually exist in much more complicated forms, yet it captures key characteristics. First, our environment reflects the rationality of customers. Customers are likely to choose the best service within their budget, but they can select a company that is not necessarily the best if it offers financial benefits, such as promotional coupons, discounts, or free services (Rowley, 1998; Familmaleki et al., 2015; Reimers & Shiller, 2019). Such a user selection represents that a user can prioritize financial advantages and change her selection, which has not been considered in the ML literature. Second, our environment realistically models a company's data acquisition. Competing companies strive to attract more users, constantly purchasing user data to improve their ML predictions. Since the data buying process could be costly for the companies, data should be carefully chosen, and this is why we incorporate AL algorithms. Our environment allows companies to use AL algorithms within finite budgets and to selectively acquire user data. Third, our environment is flexible and takes into account various competition situations in practice. Note that we make no assumptions about the number of competing predictors $M$ or budgets $n_b^{(i)}$, algorithms for predictors $f^{(i)}$ or buying strategies $\pi^{(i)}$, and the user distribution $P_{X,Y}$.

**Example 1** (Auto insurance in Section 1). *$X_t$ includes the $t$-th driver's demographic information, driving or insurance claim history, and $Y_t$ is the driver's preferred insurance plan within the user's budget constraints. Each predictor $\mathcal{C}^{(i)}$ is one insurance company (e.g., State Farm, Progressive, or AllState), offering an auto*

---

**Environment 1** A competition environment with data purchase

---

**Input:** Number of competition rounds $T$; user distribution $P_{X,Y}$; number of predictors $M$; competing predictors $\mathcal{C}^{(i)} = (n_{\mathrm{s}}^{(i)}, n_{\mathrm{b}}^{(i)}, f^{(i)}, \pi^{(i)})$ for $i \in [M]$.

**Procedure:**

For all $i \in [M]$, a model $f^{(i)}$ is trained using the $n_{\mathrm{s}}^{(i)}$ seed data points

**for** $t \in [T]$ **do**

    $(X_t, Y_t)$ from $P_{X,Y}$ is drawn and a set of buyers $\mathcal{B} = \emptyset$ is initialized

    **for** $i \in [M]$ **do**

        **if** $(n_{\mathrm{b}}^{(i)} \geq 1)$ and $(\pi^{(i)}(X_t) = 1)$ **then**

            $\mathcal{B} \leftarrow \mathcal{B} \cup \{\mathcal{C}^{(i)}\}$

        **else**

            Predict $f^{(i)}(X_t)$

        **end if**

    **end for**

    **if** $|\mathcal{B}| \geq 1$ **then**

        A user selects one predictor $W_t$ from $\mathcal{B}$ uniformly at random

        $n_{\mathrm{b}}^{(W_t)} \leftarrow n_{\mathrm{b}}^{(W_t)} - 1$

    **else**

        A user selects one predictor $W_t$ based on (1)

    **end if**

    $\mathcal{C}^{(W_t)}$ receives a user label $Y_t$ and updates $f^{(W_t)}$

**end for**

---

*insurance plan $f^{(i)}(X_t)$ based on what it predicts to be most suitable for this driver. The driver chooses one company whose offered plan is the closest to $Y_t$. If a company believes that in its database there are infrequent data from a particular group of drivers $t$-th driver belongs to (e.g., new drivers in their 30s), it can attempt to collect more data. Accordingly, the company offers discounts to attract her, and the acquired data is used to improve the company's future ML predictions.*

## 3 Experiments

Using the proposed Environment 1, we investigate the impacts of the data purchase in ML competition. Our experiments show an interesting phenomenon that data purchase can decrease the quality of the predictor selected by a user, even when the quality of the predictors gets improved on average (Section 3.1). We demonstrate that data purchase makes ML predictors similar to each other. Data purchase reduces the effective variety of options, and predictors can avoid specializing to a small subset of the population (Section 3.2). Lastly, we show the robustness of our findings against different modeling assumptions (Section 3.3).

**Metrics** To quantitatively measure the effects of data purchase, we introduce three population-level evaluation metrics. First, we define the overall quality as follows.

$$\mathbb{E}\left[\frac{1}{M} \sum_{j=1}^{M} q\left(Y, f^{(j)}(X)\right)\right], \qquad \text{(Overall quality)}$$

where the expectation is taken over the user distribution $P_{X,Y}$. The overall quality represents the average quality that competing predictors provide in the market. Second is the quality of experience (QoE), the quality of the predictor selected by a user. The QoE is defined as

$$\mathbb{E}\left[q\left(Y, f^{(W)}(X)\right)\right]. \qquad \text{(QoE)}$$

Here, the expectation is over the random variables $(X, Y, W)$ and a conditional distribution of a selected index $P(W \mid X, Y)$ is considered as Equation (1). Given that a user selects one predictor based on Equation

(1) when there is no buyer, QoE can be considered as the key utility of users. We exclude the data purchase procedure from the definition of QoE. The main reason for this exclusion is to clearly capture the user's expected satisfaction driven only by user selections after $T$ competition rounds.

Next, we define the diversity to quantify how variable the ML predictions are. To be more specific, for $i \in \mathcal{Y}$, we define the proportion of predictors whose prediction is $i$ as $p_i(X) := \frac{1}{M} \sum_{j=1}^{M} \mathbb{1}(f^{(j)}(X) = i)$. Then the diversity is defined as

$$
\mathbb{E}\left[ -\sum_{i \in \mathcal{Y}} p_i(X) \log(p_i(X)) \right], \tag{Diversity}
$$

where the expectation is taken over the marginal distribution $P_X$ and we use the convention $0 \log(0) = 0$ when $p_i(X) = 0$. Note that the diversity is defined as the expected Shannon's entropy of competing ML predictions. When there are various different options that a user can choose from, the diversity is more likely to be large.

**Implementation protocol**  Our experiments consider the seven real datasets to describe various user distributions $P_{X,Y}$, namely `Insurance` (Van Der Putten & van Someren, 2000), `Adult` (Dua & Graff, 2017), `Postures` (Gardner et al., 2014), `Skin-nonskin` (Chang & Lin, 2011), `MNIST` (LeCun et al., 2010), `Fashion-MNIST` (Xiao et al., 2017), and `CIFAR10` (Krizhevsky et al., 2009) datasets. To minimize the variance caused by other factors, we first consider a homogeneous setting in Sections 3.1 and 3.2: for each competition, all predictors have the same number of seed data $n_s^{(i)}$ and budgets $n_b^{(i)}$, the same classification algorithm for $f^{(i)}$, and the same AL algorithm for $\pi^{(i)}$. As for heterogeneous settings in Section 3.3, competitors are allowed to have different configurations of parameters.

Throughout the experiments, the total number of competition rounds is $T = 10^4$, the number of predictors is $M = 18$, and a quality function is the correctness function, *i.e.*, $q(Y_1, Y_2) = \mathbb{1}(\{Y_1 = Y_2\})$ for all $Y_1, Y_2 \in \mathcal{Y}$. We set the number for seed data points $n_s^{(i)}$ between 50 and 200 depending on the user dataset. We use either a logistic model or a neural network model with one hidden layer for $f^{(i)}$. As for the buying policy, we use a standard entropy-based AL rule for $\pi^{(i)}$ (Settles & Craven, 2008). We consider various competition situations by varying the budget $n_b \in \{0, 100, 200, 400\}$[1] and the temperature parameter $\alpha \in \{0, 1, 2, 4\}$. Note that a pair $(n_b, \alpha)$ generates one competition environment. We repeat experiments 30 times to obtain stable estimates for each pair $(n_b, \alpha)$. We provide the full implementation details in Appendix A.

To clearly capture the effect of the data purchase at a certain competition round, the data purchase procedures are not performed when we evaluate metrics. Since the evaluation metrics are defined as the population-level quantity, it is difficult to compute the expectation exactly. To handle this problem, we consider the sample averages using the i.i.d. held-out test data that are not used during the competition rounds.

### 3.1  Effects of data purchase on quality

We first study how data purchase affects the overall quality and the QoE in various competition settings. As Figure 2 illustrates, data purchase increases the overall quality as $n_b$ increases across all datasets. For instance, when $\alpha = 4$ and the dataset is `Postures`, the overall quality is 0.405 on average when $n_b = 0$, but it increases to 0.440 and 0.464 when $n_b = 200$ and $n_b = 400$, which correspond to 9% and 14% increases, respectively. As for the QoE, however, data purchase mostly decreases QoE as $n_b$ increases. For example, when the user distribution is `Insurance` and $\alpha = 2$, QoE is 0.875 when $n_b = 0$, but it reduces to 0.867 and 0.814 when $n_b = 200$ and $n_b = 400$, which correspond to 1% and 7% reduction, respectively. For `MNIST` or `Fashion-MNIST`, although there are small increases when $\alpha = 1$, QoE decreases when $\alpha = 4$.

This can be explained as follows. Given that an ML predictor attempts to collect user data when its prediction is highly uncertain, this active data acquisition increases the predictability of the individual model and reduces the model's uncertainty. Similar to AL, data purchase effectively increases a model's predictability, and so does the overall quality.

---

[1]In Section 3, for notational convenience, we often suppress the predictor index in the superscript if the context is clear. For example, we use $n_b$ instead of $n_b^{(i)}$.

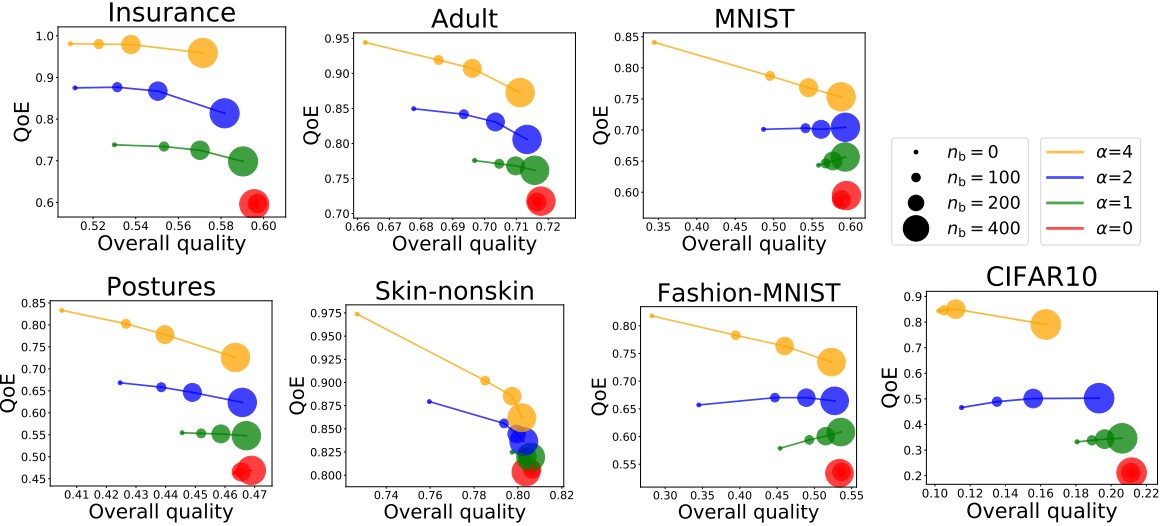

Figure 2: Illustrations of QoE as a function of the overall quality in various levels of $n_{\mathrm{b}} \in \{0, 100, 200, 400\}$ and $\alpha \in \{0, 1, 2, 4\}$ on the seven datasets. Different color indicates different $\alpha$, and the size of point indicates budgets $n_{\mathrm{b}}$. The larger budget is, the larger the point size is. In several settings, the overall quality increases as more budgets are used, but QoE decreases.

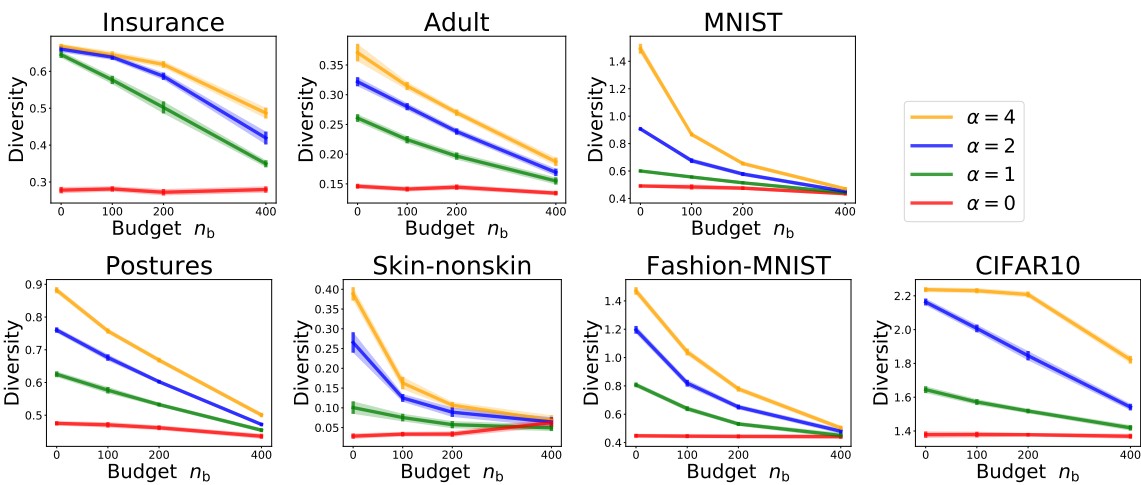

Figure 3: Illustrations of the diversity as a function of the budget $n_{\mathrm{b}}$ for various $\alpha \in \{0, 1, 2, 4\}$ on the seven datasets. Each color indicates different $\alpha$. We denote a 99% confidence band based on 30 independent runs. Competing ML predictors become similar in the sense that the diversity decreases as the budget increases.

In most cases, surprisingly, QoE decreases even when the overall quality increases. In other words, the quality that competing predictors provide is generally improved, but it does not necessarily mean that users will be more satisfied with the ML predictions. Although this result might sound counterintuitive, we demonstrate that it happens when the data purchase makes users experience fewer options, increasing the probability of finding low-quality predictions. To verify our hypothesis, we examine how data purchase affects diversity in the next section.

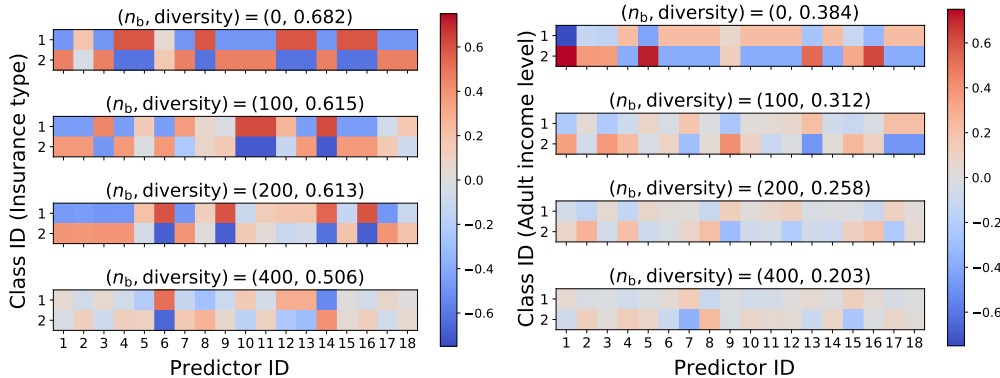

Figure 4: Heatmaps of $Q(j, y) - Q_{\text{avg}}(y)$ for (left) `Insurance` and (right) `Adult` datasets. We consider $n_{\text{b}} \in \{0, 100, 200, 400\}$ and $\alpha = 4$. The heatmaps in each row represent different $n_{\text{b}}$ but share the same color scale. For each heatmap, a horizontal axis indicates a predictor ID in $\{1, \ldots, 18\}$ and a vertical axis indicates a class in $\mathcal{Y} = \{1, 2\}$. The grid colored red (*resp.* blue) indicates a class-specific quality is higher (*resp.* lower) than average, and the white grid indicates the average. As the budget increases, the diversity decreases, and predictors produce similar class-specific quality.

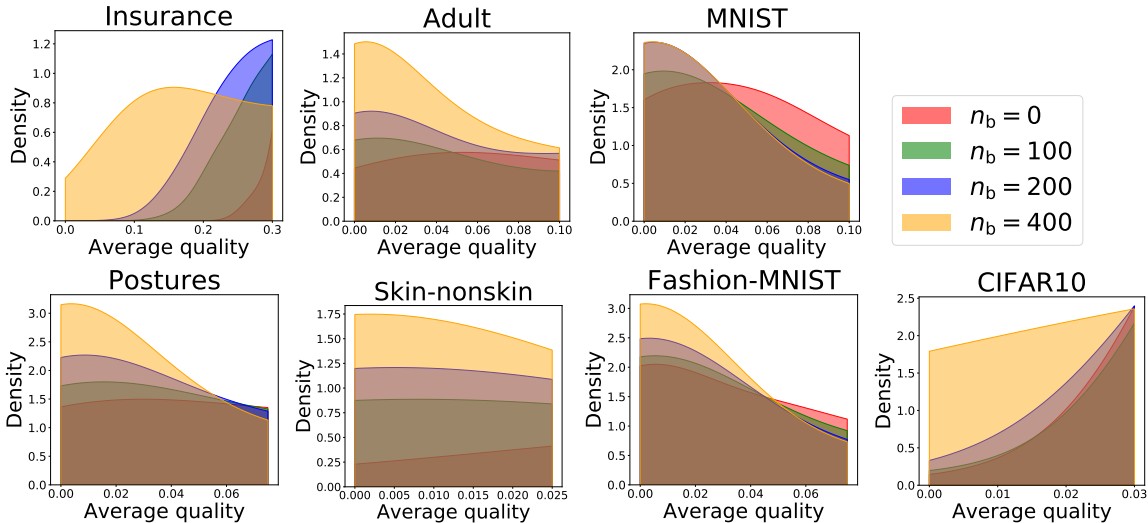

Figure 5: Probability density plots of the average quality $\frac{1}{M} \sum_{j=1}^{M} q(Y, f^{(j)}(X))$ at near zero when $n_{\text{b}} \in \{0, 100, 200, 400\}$ and $\alpha = 4$. Different color indicates different $n_{\text{b}}$. As $n_{\text{b}}$ increases, the average quality is more likely to be close to zero. That is, the probability that all ML predictors produce low-quality prediction at the same time increases, and users might not be satisfied with the ML predictions after the competing predictors purchase data.

## 3.2 Effects of data purchase on diversity

We investigate the effect of data purchase on the diversity. Figure 3 illustrates the diversity as a function of the budget $n_{\text{b}}$ in various competition settings. In general, the diversity monotonically decreases as $n_{\text{b}}$ increases across all datasets. That is, the competing predictors get similar as more budgets are allowed, and users get essentially fewer options when $n_{\text{b}}$ increases. In particular, when $\alpha = 4$ and the dataset is `Adult`, the diversity is 0.371 on average when $n_{\text{b}} = 0$, but it reduces to 0.270 and 0.187, which correspond to 27% and 50% reduction, when $n_{\text{b}} = 200$ and $n_{\text{b}} = 400$, respectively.

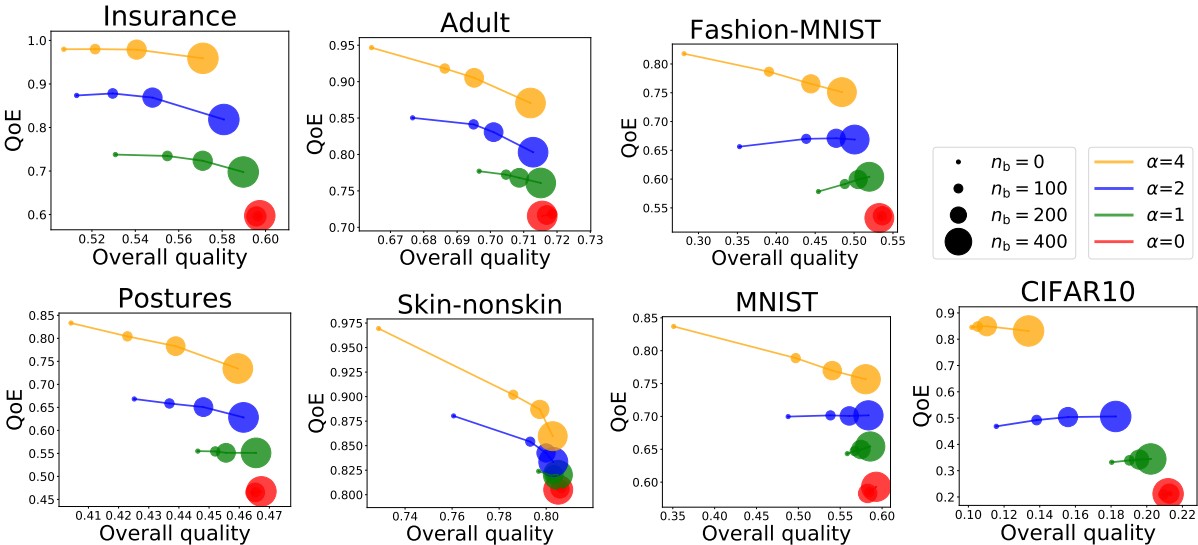

Figure 6: **Heterogeneous predictors.** Illustrations of QoE as a function of the overall quality when ML predictors have different buying strategies $\pi$. Different color indicates different $\alpha$, and the size of point indicates budgets $n_b$. The larger budget is, the larger the point size is. In several settings, the overall quality increases as more budgets are used, but QoE decreases.

We also compare the class-specific qualities of competing predictors. In Figure 4, we illustrate heatmaps of the difference $Q(j, y) - Q_{\text{avg}}(y)$ where $Q(j, y)$ is the class-specific quality defined as $Q(j, y) :=$ $\mathbb{E}\left[q\left(Y, f^{(j)}(X)\right) \mid Y = y\right]$ for $j \in [M]$ and $y \in \mathcal{Y}$, and $Q_{\text{avg}}(y)$ for $y \in \mathcal{Y}$ is defined as $\frac{1}{M} \sum_{j=1}^{M} Q(j, y)$. This difference measures the class-specific quality of a company, showing how specialized its ML model is. We use the `Insurance` and `Adult` datasets. When $n_b = 0$, the `Adult` heatmap shows that `predictor 1` and `predictor 5` so specialize to `class 2` prediction that they sacrifice their prediction power for `class 1` compared to other predictors. In other words, competition rounds encourage each ML model to be very specialized in a small subgroup of the population. However, when $n_b = 400$, all predictors have similar levels of class-specific quality. The data purchase makes competing ML predictors similar and helps predictors not too much focus on a subgroup of the population. Similar results are shown in Ginart et al. (2021), but one finding that the previous works have not shown is that this specialization can be alleviated when predictors purchase user data.

**Implications of decreases in diversity** We now examine the connection between the diversity and the quality of the predictor selected by a user. We demonstrate that the probability of finding low-quality predictions can increase due to the reduction in diversity, explaining how diversity affects QoE. Figure 5 illustrates the probability density functions of the average quality near zero. It clearly shows that the probability that the average quality is near zero increases as more budgets $n_b$ are used: the areas for $n_b = 400$ (colored in yellow) are clearly larger than those for $n_b = 0$ (colored in red). That is, as predictions become similar, it is more likely that all ML predictions have a low quality simultaneously. Hence, the probability that users are not satisfied with the predictions increases, and leading to decreases in QoE even when the overall quality increases.

### 3.3 Robustness to modeling assumptions

We further demonstrate that our findings are robust against different modeling assumptions. We consider the same situation used in the previous sections but ML predictors now can have different buying strategies $\pi$. To be more specific, we consider the three different types of buying strategies by varying the threshold of the uncertainty-based AL method. For $C_{\text{Ent}} \in \{0, 0.3, 0.6\}$, we consider the following buying strategy models

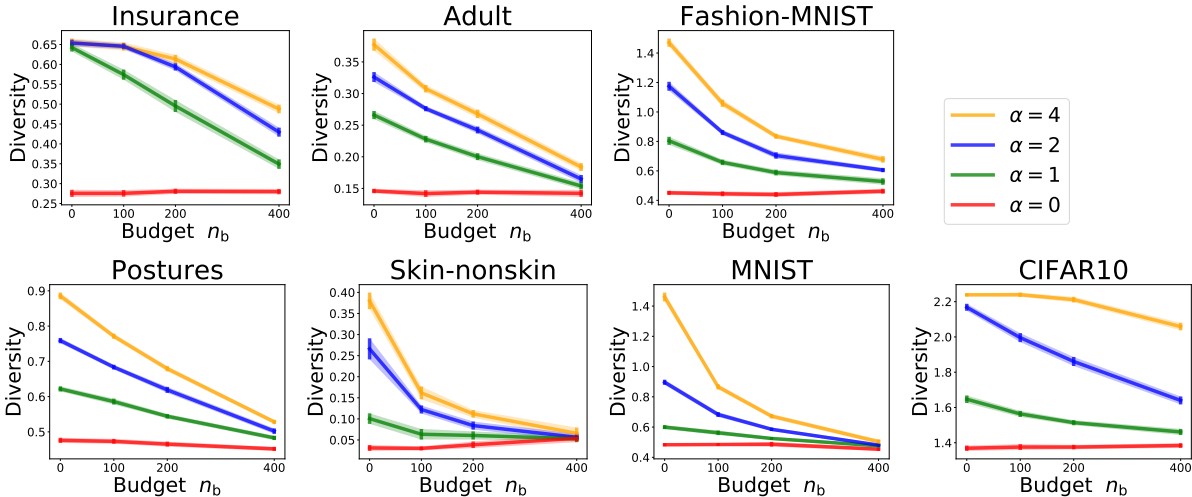

Figure 7: **Heterogeneous predictors.** Illustrations of the diversity as a function of the budget $n_b$ when ML predictors have different buying strategies $\pi$. Different color indicates different $\alpha$. We denote a 99% confidence band based on 30 independent runs. As the budget increases, the diversity decreases.

$\pi_{C_{\mathrm{Ent}}}(X_t) = \mathbb{1}(\{\mathrm{Ent}(p^{(i)}(X_t)) \geq C_{\mathrm{Ent}} \log(|\mathcal{Y}|)\})$, where Ent is the Shannon's entropy function and $p^{(i)}(X_t)$ is the probability estimate given $X_t$. We assume there are 6 predictors for each buying strategy $\pi_0$, $\pi_{0.3}$, and $\pi_{0.6}$. This modeling assumption considers the situation where there are three groups with different levels of sensitivity to data purchases. For instance, in our setting, $\pi_{0.6}$ is the most conservative data buyer and is less likely to buy new data.

Figure 6 shows the relationship between the QoE and the overall quality when there are heterogeneous ML predictors with different buying strategies. Similar to Figure 2, the overall quality increases but QoE generally decreases as the budget increases across all datasets. As for the diversity, as anticipated, Figure 7 shows that the diversity decreases as the budget increases. It demonstrates that our findings are robustly observed against different environment settings. We also conduct more experiments (i) when budgets $n_b$ are different across companies or (ii) when there are different number of predictors $M$. All these additional results are provided in Appendix B.

## 4 Theoretical analysis on competition

In this section, we establish a simple representation for QoE when a quality function is the correctness function. Based on this finding, we theoretically analyze how the diversity-like quantity can affect QoE. Proofs are provided in Appendix D.

**Lemma 1** (A simple representation for QoE). *Suppose there is a set of $M \geq 2$ predictors $\{f^{(j)}\}_{j=1}^{M}$ and a quality function is the correctness function, i.e., $q(Y_1, Y_2) = \mathbb{1}(\{Y_1 = Y_2\})$ for all $Y_1, Y_2 \in \mathcal{Y}$. Let $Z := \frac{1}{M} \sum_{j=1}^{M} q\left(Y, f^{(j)}(X)\right)$ be the average quality for a user $(X, Y)$. For any $\alpha \geq 0$, we have*

$$\mathbb{E}[Z] = (\text{Overall quality}) \leq (\text{QoE}) = \mathbb{E}\left[\frac{Ze^{\alpha}}{Ze^{\alpha} + (1 - Z)}\right], \tag{2}$$

*where the inequality holds when $\alpha = 0$ and the expectation is considered over $P_{X,Y}$.*

Lemma 1 presents a relationship between QoE and the overall quality—QoE is always greater than the overall quality if $\alpha > 0$. In addition, it shows that QoE can be simplified as a function of the average quality $Z$ over competitors when a quality function $q$ is the correctness function. Using the relationship shown in Lemma 1, we provide a sufficient condition for the overall quality to be greater but the QoE to be less in the following theorem.

**Theorem 1** (Comparison of two competition dynamics). *Suppose there are two sets of $M \geq 2$ predictors, $\mathcal{F}_1 := \{f^{(j)}\}_{j=1}^M$ and $\mathcal{F}_2 := \{g^{(j)}\}_{j=1}^M$. Without loss of generality, the overall quality for $\mathcal{F}_2$ is larger than that of $\mathcal{F}_1$. For the correctness function $q$, we define $Z_1 := \frac{1}{M}\sum_{j=1}^M q(Y, f^{(j)}(X))$ and $Z_2 := \frac{1}{M}\sum_{j=1}^M q(Y, g^{(j)}(X))$ as Lemma 1. If $\alpha \geq C_\alpha$ and $\mathrm{Var}[Z_2] \geq C_1 \mathrm{Var}[Z_1]$ for some explicit constants $C_\alpha$ and $C_1 \leq 1$, then QoE for $\mathcal{F}_2$ is smaller than that for $\mathcal{F}_1$.*

Theorem 1 compares two competition dynamics, $\mathcal{F}_1$ and $\mathcal{F}_2$, providing a sufficient condition for when QoE for $\mathcal{F}_2$ is smaller than that for $\mathcal{F}_1$ whereas the associated overall quality is larger. Theorem 1 implies that QoE can decrease when $\mathrm{Var}[Z_2]$ is large enough compared to $\mathrm{Var}[Z_1]$. Considering our results in Figures 3, 4, and 5 that data purchase makes competing predictors similar when $\alpha$ is large enough, the average quality is more likely to become zero or one. As a result, it increases variance $\mathrm{Var}[Z_2]$ because the variance is maximized when random variables spread over $[0,1]$. To be more specific, on `Insurance`, we compare two environments where one consists of competitors with $n_\mathrm{b} = 0$, and the other has competitors with $n_\mathrm{b} = 400$ when $\alpha = 4$. We denote a set of competing predictors by $\mathcal{F}_1$ when $n_\mathrm{b} = 0$ and $\mathcal{F}_2$ when $n_\mathrm{b} = 400$. In this setting, the constants are $(C_\alpha, C_1) = (0.03, 0.08)$ and the variances are $(\mathrm{Var}[Z_1], \mathrm{Var}[Z_2]) = (0.01, 0.11)$, making the inequality assumptions in Theorem 1 feasible.

Ginart et al. (2021) theoretically examines how the number of competitors affects QoE, showing having too few or too many competitors may decrease the QoE. Our result in Theorem 1 focuses on the relationship between the overall accuracy and QoE, explaining why QoE decreases when ML predictors can actively acquire user data through the data purchase system, supporting our main findings in experiments.

### 4.1 QoE for a general quality function

When the quality function $q$ is not the correctness function, QoE does not have a comprehensible representation as in Lemma 1. The following theorem shows the upper and lower bounds of QoE for a general quality function.

**Theorem 2.** *Suppose there is a set of $M \geq 2$ prediction models $\{f^{(j)}\}_{j=1}^M$. For any non-negative function $q: \mathcal{Y} \times \mathcal{Y} \to \mathbb{R}_+$ and $\alpha \geq 0$, we have the following upper and lower bounds.*

$$(\textit{Overall quality}) \leq \mathrm{QoE} \leq \mathbb{E}\left[\max_{j \in [M]} q\left(Y, f^{(j)}(X)\right)\right].$$

*The equality for the left and right equations holds when $\alpha = 0$ and $\alpha = \infty$, respectively.*

Theorem 2 shows that QoE is lower bounded by the overall quality as in Lemma 1 and is upper bounded by the expectation of maximum quality when $\alpha = \infty$. In other words, QoE is the highest quality of predictions available in the market when users have strong confidence in which predictions are best. The impact of different choices of quality functions on MQ, QoE, or the relation between them is an interesting research problem, but we leave it to a future topic.

## 5 Conclusion

In this paper, characterizing the nature of competition and data purchase, we propose a new competition environment in which ML predictors are allowed to actively acquire labeled user data and improve their models. Our results show that even though the data purchase improves the quality that predictors provide, it can decrease the quality that users experience. We explain this counterintuitive finding by demonstrating that data purchase makes competing predictors similar to each other in various situations.

In order to derive tractable analysis and experiments, we have to make some modeling simplifications. Similar simplifications are commonly used in ML and multi-agent literature, and these are necessary here, especially because there lacks systematic analysis of data purchase in competition. For example, one assumption in our environment for simplicity is that the user distribution does not change over time. In practice, customer behavior can change or evolve over time (Jin & Vasserman, 2019; Reimers & Shiller, 2019). Another assumption we used on the competitor side is that the ML competitor's purchase intent is dichotomous:

either buy or not. In practice, willingness is often described with continuous values, as ML competitors can pay the price as much as they want, even affecting how users select a service provider. Hence, it is important to expand various directions of our research in future works.

## Broader impact statement

Our findings can broadly benefit the ML communities by providing insights into how competition over datasets and data acquisitions can affect a user's experiences. As more companies shift their focus to data science and AI, there will be increasing competition in acquiring valuable data. We believe it is important to investigate the impact and potential biases that such competition induce in the machine learning training and predictions.

## Acknowledgments

The authors would like to thank all anonymous reviewers for their constructive comments.

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

## Appendix

In this appendix, we provide implementation details in Appendix A, additional numerical experiments in Appendix B, and proofs and additional theoretical results in Appendix D.

## A    Implementation details

In this section, we provide implementation details. We explain the user distribution, ML predictors, and the proposed environment. Our Python-based implementations are available at `https://github.com/ykwon0407/data_purchase_in_comp`.

**Datasets (user distribution)**  As for the datasets (user distribution $P_{X,Y}$), we used the following seven datasets for our experiments: `Insurance` (Van Der Putten & van Someren, 2000), `Adult` (Dua & Graff, 2017), `Postures` (Gardner et al., 2014), `Skin-nonskin` (Chang & Lin, 2011), `FashionMNIST` (Xiao et al., 2017), `MNIST` (LeCun et al., 2010), and `CIFAR10` (Krizhevsky et al., 2009). For all datasets, we first split a dataset into competition and evaluation datasets: the competition dataset is used during the $T = 10^4$ competition rounds and the evaluation dataset is used for evaluate metrics after the competition. For `FashionMNIST`, `MNIST`, and `CIFAR10`, we use the original training and test datasets for competition and evaluation datasets, respectively. For `Insurance`, `Adult`, `Postures`, and `Skin-nonskin`, we randomly sample 5000 data points from the original dataset to make the evaluation dataset and use the remaining data points as the competition dataset. To account for uncertainty in user selection, user labels are randomly flipped for 30% of user data points. At each round of competition, we randomly sample one data point from the competition dataset. After the $T$ competition rounds, we randomly sample 3000 points from the evaluation dataset and evaluate the metrics (the overall quality, QoE, and diversity). Note that all of experiment results are based on the evaluation dataset. Table 1 shows a summary of the seven datasets used in our experiments.

Table 1: A summary of datasets used in our experiments.

| Dataset | The size of competition dataset | The size of evaluation dataset | Input dimension | # of classes $|\mathcal{Y}|$ |
|---|---|---|---|---|
| Insurance | 13823 | 5000 | 16 | 2 |
| Adult | 43842 | 5000 | 108 | 2 |
| Postures | 69975 | 5000 | 15 | 5 |
| Skin-nonskin | 239057 | 5000 | 3 | 2 |
| Fashion-MNIST | 60000 | 10000 | 784 | 10 |
| MNIST | 60000 | 10000 | 784 | 10 |
| CIFAR10 | 50000 | 10000 | 3072 | 10 |

As for the preprocessing, we apply the standardization to have zero mean and one standard deviation for `Skin-nonskin`. For the two image datasets, `MNIST` and `CIFAR10` we apply the channel-wise standardization.

Other than the three datasets, we do not apply any other preprocessing. To reflect the customers' randomness in their selection, we apply a random noise on the original label. We assign a random label with 30% for every dataset. This random perturbation is applied to both the competition and evaluation datasets.

**ML predictors** We fix the number of predictors to $M = 18$ throughout our experiments. For each dataset, which makes one competition environment, we consider a homogeneous setting, *i.e.*, all predictors have the same number of seed data $n_s^{(i)}$, a budget $n_b^{(i)}$, a model $f^{(i)}$, and a buying strategy $\pi^{(i)}$. As for the buying strategy, we fix $\pi^{(i)}(X_t) = \mathbb{1}(\{\text{Ent}(p^{(i)}(X_t)) \geq 0.3 \log(|\mathcal{Y}|)\})$, where $\text{Ent}(p^{(i)}(X_t))$ is the Shannon's entropy of $p^{(i)}(X_t)$, and $p^{(i)}(X_t)$ is the corresponding probability estimate for $P(Y = Y_t)$. That is, if the entropy is higher than the pre-defined threshold $0.3 \log(|\mathcal{Y}|)$, a predictor decides to buy the user data. Note that $\log(|\mathcal{Y}|)$ is the Shannon's entropy of the uniform distribution on $\mathcal{Y}$.

Table 2 shows a summary information for the seed data $n_s$ and the model $f$ for each dataset. Every ML predictor initially trains with the $n_s$ seed data points. For all experiments, we use the Adam optimization (Kingma & Ba, 2014) with the specified learning rate and epochs. The batch size is fixed to 64. If a predictor is selected, then its ML model is updated with one iteration with the newly obtained data point, and we retrain the model whenever the 'retrain period' new samples are obtained.

Table 2: A summary of hyperparameters by datasets. Logistic denotes a logistic model and NN denotes a neural network one hidden layer.

| Dataset | Seed data $n_s$ | ML predictor $f$ | | | | |
|---|---|---|---|---|---|---|
| | | Model | # of hidden nodes | Epoch | Learning rate | Retrain period |
| Insurance | 100 | Logistic | - | 10 | $5 \times 10^{-3}$ | 50 |
| Adult | 100 | Logistic | - | 10 | $10^{-2}$ | 50 |
| Postures | 200 | Logistic | - | 10 | $3 \times 10^{-2}$ | 50 |
| Skin-nonskin | 50 | Logistic | - | 10 | $3 \times 10^{-2}$ | 50 |
| Fashion-MNIST | 50 | NN | 400 | 30 | $10^{-4}$ | 150 |
| MNIST | 50 | NN | 400 | 30 | $10^{-4}$ | 150 |
| CIFAR10 | 100 | NN | 400 | 30 | $10^{-4}$ | 150 |

# B   Additional numerical experiments

In this section, we provide additional experimental results to demonstrate the robustness of our findings against different modeling assumptions in the heterogeneous setting. As for the heterogeneous settings, we consider different budgets in the subsection B.1 and different number of competing predictors in the subsection B.2. All additional results again show the robustness of our experimental findings against different modeling assumptions.

## B.1   Different budgets

We use the same setting used in the homogeneous setting but with different budgets. We use the `Insurance`, `Adult`, and `Skin-nonskin` datasets. For $n_b^{(i)} \in \{0, 100, 200, 400\}$, we assume that the first 9 predictors have $n_b^{(i)}$ budgets, but the last 9 predictors have $n_b^{(i)}/2$ budgets. That is, half of the predictors have half the budget compared to the other group. This situation can be considered as some groups of companies have a larger amount of capital than others. Figure 8 shows that the main findings appear again even when different budgets are used (QoE generally decreases, overall quality increases, and diversity decreases).

## B.2   Different number of competing predictors

We also show that our findings are consistent for the different number of competing predictors in the market. All the experiments in Section 3 of the manuscript consider $M = 18$. Here, we consider the homogeneous

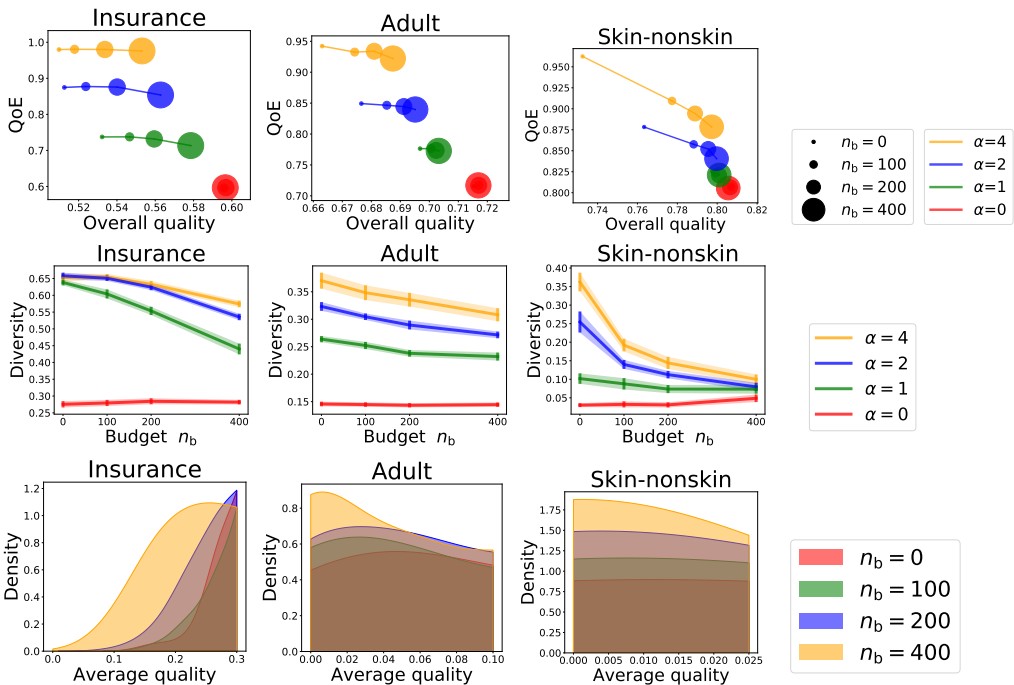

Figure 8: **Heterogeneous predictors.** Main figures when competing predictors use different budgets. The results are similar to the homogeneous setting, showing the robustness of our main findings.

setting but a different number of competing predictors $M = 9$ or $M = 12$. As Figures 9 and 10 show, the main findings are captured again when the number of predictors are used.

### B.3 Different active learning algorithms

We conduct additional experiments with different active learning algorithms. Specifically, we consider confidence-based active learning algorithms, *i.e.*, competitors decide to buy a user datum based on the confidence of their model prediction. Specifically, the confidence we considered is the maximum predicted probabilities over different classes:

$$\pi^{(i)}(X_t) = \mathbb{1}(\{\max_{j \in \mathcal{Y}}(p_j^{(i)}(X_t)) \leq 1.2/|\mathcal{Y}|\}). \tag{3}$$

Given that $\max_{j \in \mathcal{Y}}(p_j^{(i)}(X_t))$ is always bigger than $1/|\mathcal{Y}|$, our confidence-based active learning algorithm in (3) will try to purchase user data if a predicted probability vector is close to the random guess. As Figure B.3 shows on two datasets (`Skin-nonskin` and `Adult`), our main findings are consistently observed even when competing predictors use different active learning algorithms for data purchase.

## C  Consistent results from relaxing model assumptions

In this section, we relax the model assumptions to make proposed competition environments more realistic. We consider the following two extensions: (i) each user datum has a different value and (ii) a user chooses one competitor based on their prediction quality even when there is more than one buyer.

The setting (i) is to consider the case where some user data can be cleaner or noisier than others, and accordingly, some data values are more expensive or cheaper than others. Specifically, we divide users into two groups: a clean group and a noisy group. We assign a user to the noisy groups if its label is changed from the random flip, and the rest are in the clean group. Note that labels have been randomly flipped 30% of users in the homogeneous setting. We set the data prices for the clean and noisy groups are 8/7 and 2/3,

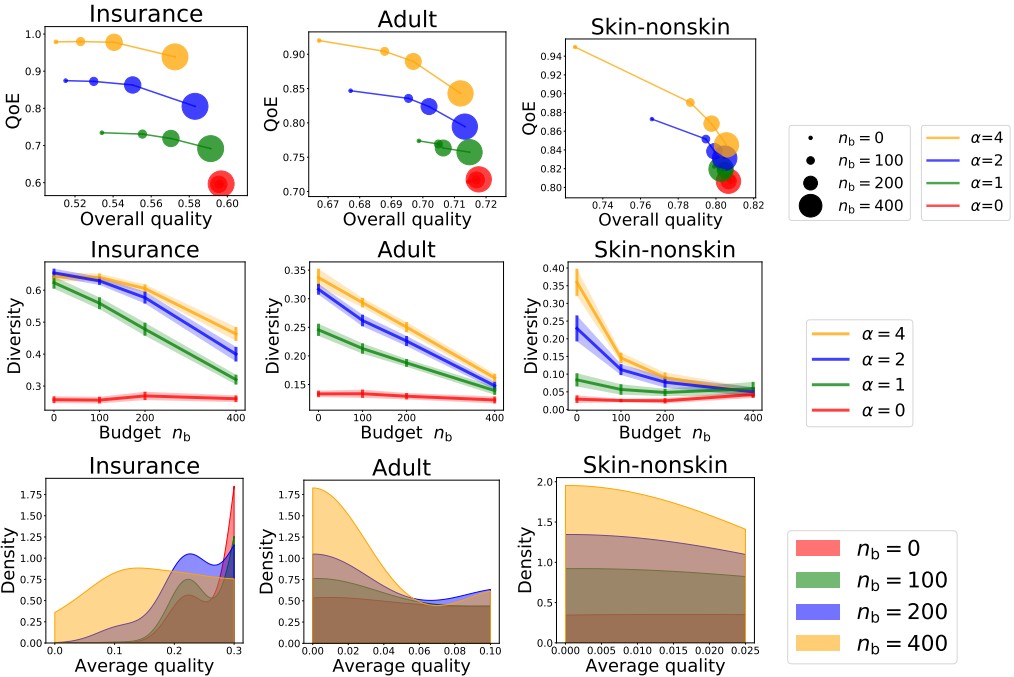

Figure 9: **Heterogeneous predictors.** Main figures when there are $M = 9$ competing predictors. The results are similar to the $M = 18$, showing the robustness of our main results.

respectively. This choice of data price does not change the total amount of the price from the homogeneous setting.

The setting (ii) is to consider the setting where a user prefers a buyer with a high-quality model prediction. Specifically, we consider the Equation (1) but with buyers instead of selecting one uniformly at random. At $t$-th round, for a set of buyers $\mathcal{B}_t \subseteq [M]$ and $i \in \mathcal{B}_t$,

$$P\left(W_t = i \mid Y_t, \{f^{(j)}(X_t)\}_{j \in \mathcal{B}_t}\right) = \frac{\exp\left(\alpha q\left(Y_t, f^{(i)}(X_t)\right)\right)}{\sum_{j \in \mathcal{B}_t} \exp\left(\alpha q\left(Y_t, f^{(j)}(X_t)\right)\right)}.$$

Figures 12 and 13 show the main figures for the settings (i) and (ii), respectively. We consider `Skin-nonskin` and `Adult` datasets for user data distributions. Experimental results show that our main findings are consistent under these relaxations—i.e., the prediction quality users experience can decrease even when the overall quality increases.

## D  Proofs and additional theoretical results

We provide proofs for Lemma 1 and Theorem 1 in the subsection D.1.

### D.1  Proofs

*Proof of Lemma 1.* For notational convenience, we set $q^{(j)} := q(Y, f^{(j)}(X))$ for $j \in [M]$.

$$\mathbb{E}\left[q\left(Y, f^{(W(\alpha))}(X)\right)\right] = \mathbb{E}\left[\mathbb{E}\left[q\left(Y, f^{(W(\alpha))}(X)\right) \mid Y, \{f^{(j)}\}_{j=1}^M\right]\right] = \mathbb{E}\left[\sum_{j=1}^M p^{(j)}(\alpha)q^{(j)}\right], \tag{4}$$

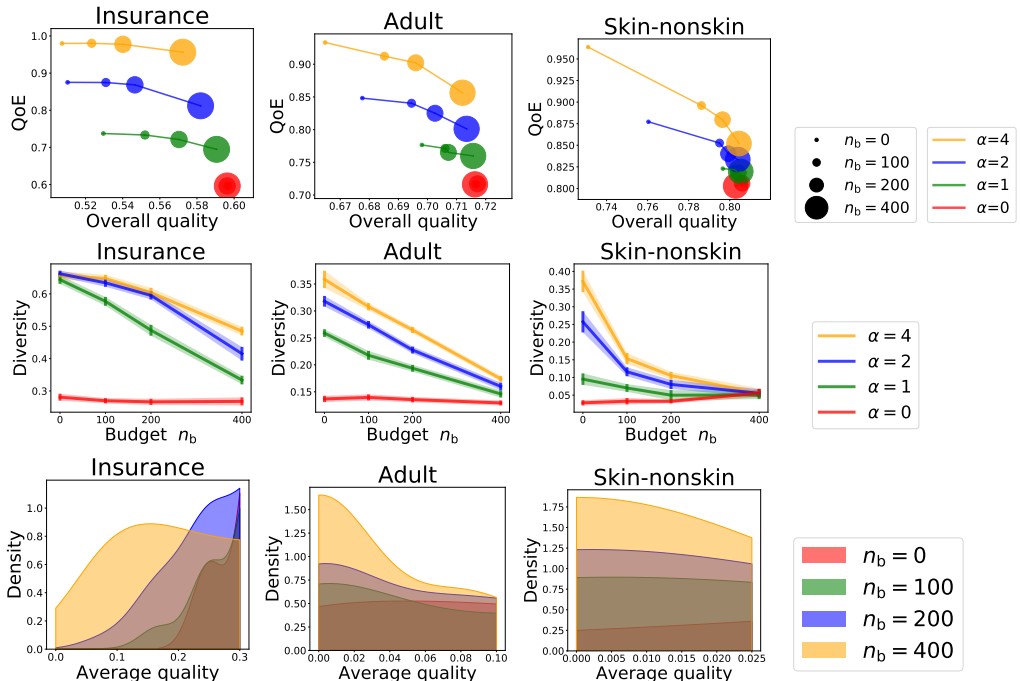

Figure 10: **Heterogeneous predictors.** Main figures when there are $M = 12$ competing predictors. The results are similar to the $M = 18$, showing the robustness of our main results.

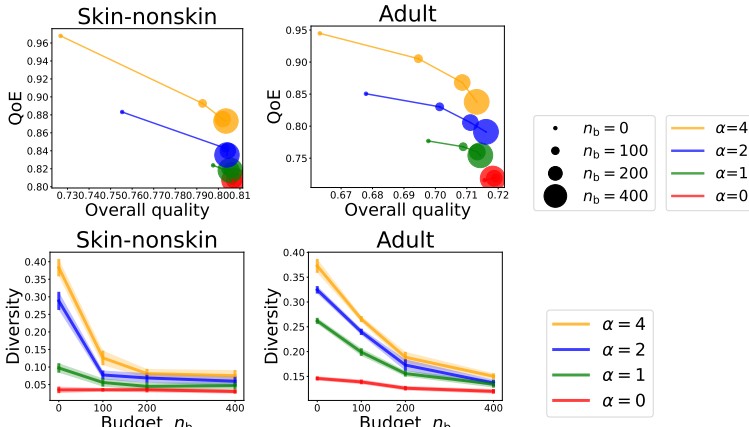

Figure 11: **Different active learning algorithms.** Main figures when competing predictors use the confidence-based active learning algorithm for data purchase. The results are similar to the uncertainty-based active learning algorithm cases, showing the robustness of our main results.

where for $i \in [M]$,

$$p^{(i)}(\alpha) = \frac{\exp\left(\alpha q^{(i)}\right)}{\sum_{j=1}^{M} \exp\left(\alpha q^{(j)}\right)}.$$

Since $q^j = \mathbb{1}(\{Y = f^{(j)}(X)\}) \in \{0, 1\}$, for $N_{\text{cor}} := \sum_{j=1}^{M} q(Y, f^{(j)}(X)) = \sum_{j=1}^{M} \mathbb{1}(\{Y = f^{(j)}(X)\})$, we have

$$p^{(j)}(\alpha) = \frac{\exp(\alpha \mathbb{1}(\{Y = f^{(j)}(X)\}))}{N_{\text{cor}} \exp(\alpha) + (M - N_{\text{cor}})},$$

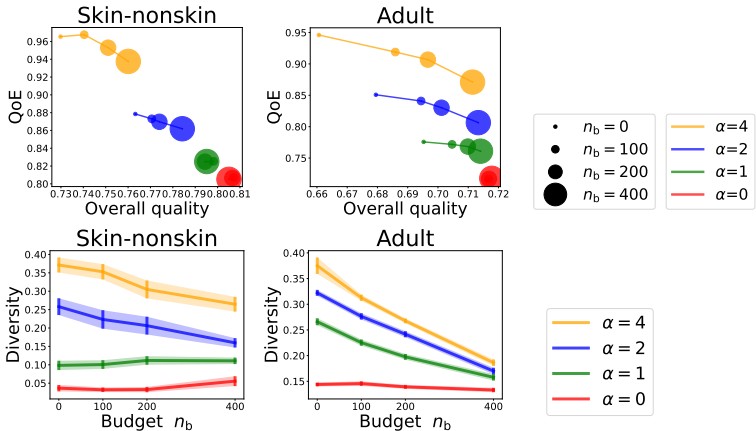

Figure 12: **Non-constant data price:** We consider the setting where the user data price is not fixed to a constant.

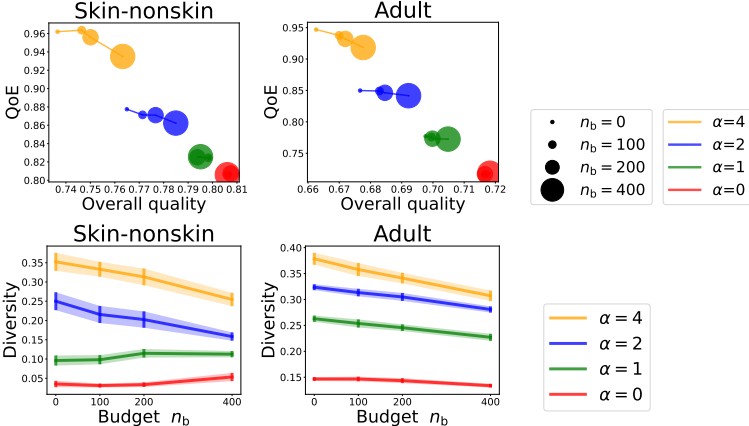

Figure 13: **Non-random selection:** In this setting, a user selects one of buyers based on their prediction quality.

and

$$\sum_{j=1}^{M} p^{(j)}(\alpha) q^{(j)} = \frac{1}{M} \sum_{j=1}^{M} \frac{e^{\alpha} \mathbb{1}(\{Y = f^{(j)}(X)\})}{(N_{\text{cor}}/M)e^{\alpha} + (1 - N_{\text{cor}}/M)} = \frac{(N_{\text{cor}}/M)e^{\alpha}}{(N_{\text{cor}}/M)e^{\alpha} + (1 - N_{\text{cor}}/M)}.$$

Since $Z = N_{\text{cor}}/M$ and $k(z, \alpha) := \frac{ze^{\alpha}}{ze^{\alpha} + (1-z)}$ is an increasing function, it concludes a proof. $\qquad \square$

*Proof of Theorem 1.* For $0 \le z \le 1$ and $\alpha \ge 0$, let $k(z, \alpha) := \frac{ze^{\alpha}}{ze^{\alpha} + (1-z)}$, $\mu_1 := \mathbb{E}[Z_1]$, and $\mu_2 := \mathbb{E}[Z_2]$. Note that

$$\mathbb{E}\left[q\left(Y, f^{(W(\alpha))(X)}\right)\right] = \mu_1 + \mathbb{E}\left[k(Z_1, \alpha) - Z_1\right]$$

$$\mathbb{E}\left[q\left(Y, g^{(W(\alpha))}(X)\right)\right] = \mu_2 + \mathbb{E}\left[k(Z_2, \alpha) - Z_2\right].$$

Thus, we have

$$\mathbb{E}\left[q\left(Y, f^{(W(\alpha))(X)}\right)\right] \ge \mathbb{E}\left[q\left(Y, g^{(W(\alpha))}(X)\right)\right]$$

$$\iff \quad \mathbb{E}\left[k(Z_1, \alpha) - Z_1\right] - \mathbb{E}\left[k(Z_2, \alpha) - Z_2\right] \ge \mu_2 - \mu_1.$$

For $Z \in \{\frac{1}{M}, \ldots, \frac{M-1}{M}\}$, we have

$$\frac{M(e^{\alpha} - 1)}{(M-1)e^{\alpha} + 1} \le \frac{e^{\alpha} - 1}{Ze^{\alpha} + (1 - Z)} \le \frac{M(e^{\alpha} - 1)}{e^{\alpha} + (M-1)}.$$

Therefore, since $k(z) - z = \frac{Z(1-Z)(e^{\alpha}-1)}{Ze^{\alpha}+(1-Z)}$, we have

$$\frac{M(e^{\alpha} - 1)}{(M-1)e^{\alpha} + 1} Z(1 - Z) \le k(Z) - Z \le \frac{M(e^{\alpha} - 1)}{e^{\alpha} + (M-1)} Z(1 - Z). \tag{5}$$

Let $C_{\text{low}} = \frac{M(e^{\alpha}-1)}{(M-1)e^{\alpha}+1}$ and $C_{\text{upp}} = \frac{M(e^{\alpha}-1)}{e^{\alpha}+(M-1)}$. From the inequalities (5), we have

$$\mathbb{E}\left[k(Z_1, \alpha) - Z_1\right] - \mathbb{E}\left[k(Z_2, \alpha) - Z_2\right] \ge C_{\text{low}} \mathbb{E}\left[Z_1(1 - Z_1)\right] - C_{\text{upp}} \mathbb{E}\left[Z_2(1 - Z_2)\right]$$
$$= C_{\text{low}}(\mu_1(1 - \mu_1) - \text{Var}[Z_1]) - C_{\text{upp}}(\mu_2(1 - \mu_2) - \text{Var}[Z_2]).$$

The last equality is due to $\mathbb{E}\left[Z(1-Z)\right] = \mathbb{E}[Z](1 - \mathbb{E}[Z]) - \text{Var}(Z)$. Therefore, QoE is decreased if

$$C_{\text{low}}(\mu_1(1 - \mu_1) - \text{Var}[Z_1]) - C_{\text{upp}}(\mu_2(1 - \mu_2) - \text{Var}[Z_2]) \ge \mu_2 - \mu_1$$

$$\iff \quad \text{Var}[Z_2] \ge \frac{C_{\text{low}}}{C_{\text{upp}}} \left(\text{Var}[Z_1] - \mu_1(1 - \mu_1)\right) + \frac{1}{C_{\text{upp}}} (\mu_2 - \mu_1) + \mu_2(1 - \mu_2)$$

$$\iff \quad \text{Var}[Z_2] \ge C_1 \text{Var}[Z_1] + C_2(\mu_1, \mu_2),$$

where

$$C_1 := \frac{C_{\text{low}}}{C_{\text{upp}}} = \frac{e^{\alpha} + (M-1)}{(M-1)e^{\alpha} + 1} \le 1$$

$$C_2(\mu_1, \mu_2) := -C_1 \mu_1(1 - \mu_1) + \frac{1}{C_{\text{upp}}} (\mu_2 - \mu_1) + \mu_2(1 - \mu_2)$$

Therefore, if there is a constant $C_{\alpha}$ such that $C_2 \ge 0$, then it concludes a proof.

By definition of $C_2$, it is positive when $\mu_2(1 - \mu_2) - C_1 \mu_1(1 - \mu_1)$.

$$\mu_2(1 - \mu_2) - C_1 \mu_1(1 - \mu_1) > 0$$

$$\iff \quad C_1 \le \frac{\mu_2(1-\mu_2)}{\mu_1(1-\mu_1)}$$

$$\iff \quad \frac{e^\alpha + (M-1)}{(M-1)e^\alpha + 1} \le \frac{\mu_2(1-\mu_2)}{\mu_1(1-\mu_1)}$$

$$\iff \quad e^\alpha \ge \frac{(M-1)\mu_1(1-\mu_1) - \mu_2(1-\mu_2)}{(M-1)\mu_2(1-\mu_2) - \mu_1(1-\mu_1)}.$$

By setting $C_\alpha = \log \frac{(M-1)\mu_1(1-\mu_1) - \mu_2(1-\mu_2)}{(M-1)\mu_2(1-\mu_2) - \mu_1(1-\mu_1)}$, it concludes a proof.

$\square$

*Proof of Theorem 2.* We use the same notations in the proof of Lemma 1. We first show QoE is an increasing function as $\alpha$. From the representation (4), we have

$$\frac{\partial \mathbb{E}\left[\sum_{j=1}^M p^{(j)}(\alpha) q^{(j)}\right]}{\partial \alpha}$$

$$= \mathbb{E}\left[\sum_{j=1}^M \frac{\partial p^{(j)}(\alpha)}{\partial \alpha} q^{(j)}\right]$$

$$= \mathbb{E}\left[\sum_{j=1}^M \frac{\left(\exp\left(\alpha q^{(j)}\right) q^{(j)} \left(\sum_{k=1}^M \exp\left(\alpha q^{(k)}\right)\right)\right) - \left(\exp\left(\alpha q^{(j)}\right) \sum_{k=1}^M \exp\left(\alpha q^{(k)}\right) q^{(k)}\right)}{\left(\sum_{j=1}^M \exp\left(\alpha q^{(j)}\right)\right)^2} q^{(j)}\right]$$

$$= \mathbb{E}\left[\sum_{j=1}^M p^{(j)}(\alpha)(q^{(j)} - \bar{q}) q^{(j)}\right],$$

where $\bar{q} := \sum_{k=1}^M p^{(k)}(\alpha) q^{(k)}$. From the last equality, we have

$$\sum_{j=1}^M p^{(j)}(\alpha)(q^{(j)} - \bar{q}) q^{(j)} = \sum_{j=1}^M p^{(j)}(\alpha)(q^{(j)})^2 - \bar{q}^2 > 0.$$

Note the non-negativity is from Cauchy-Schwarz inequality. We now prove an upper bound. Note that

$$\sum_{j=1}^M p^{(j)}(\alpha) q^{(j)} \le \max_{j \in [M]} q^{(j)},$$

and the equality holds when $\alpha = \infty$. Therefore, taking expectations on both sides provides an upper bound. As for the lower bound. Due to the representation (4), it is enough to show that

$$\sum_{j=1}^M p^{(j)}(\alpha) q^{(j)} \ge \frac{1}{M} \sum_{j=1}^M q^{(j)}.$$

Since QoE is an increasing function, by plugging in $\alpha = 0$, it gives $p^{(j)}(\alpha) = 1/M$. It concludes a proof. $\square$

