# OpenReview forum: "Competition over data: how does data purchase affect users?"
_TMLR — Accepted by TMLR_

### Review · Reviewer_WCny · 2022-08-15

**Summary Of Contributions:**

This paper studies an interesting environment in which ML predictors use active learning algorithms to effectively acquire labeled data within their budgets while competing against each other. It models the problem from two sides. On the one side, it models that M competing predictors independently train their model using some seed data points.  After initialization, each predictor determines whether to buy the user data. On the other side, the paper also models how a user selects one predictor. At each round, the user selects only one predictor based on both purchase intents and prediction information. The paper empirically shows that the overall performance of an ML predictor improves when predictors can purchase additional labeled data. It also shows a surprising result that the accuracy of the predictor selected by each user can decrease even as the individual predictors get better.

**Broader Impact Concerns:**

I hope authors can also discuss the fairness issue that might be involved in this problem from both the predictor and customer's side.

**Requested Changes:**

1. Problem setup:

- It assumes that all labeled data would be worth one unit of budget. I think this assumption is a bit restrictive. Different data may have different values, especially when some data is much more difficult to obtain. For example, some products are much more popular than others so the data should be cheaper to purchase.

- The modeling part on the predictor side is a little bit weak.
   To model the problem more realistically, I think it is natural to let the predictors decide how much they are willing to pay for the data. Then the willingness of the user selecting a particular predictor not only depends on how good the predictor is, but also the budget that they pay for. This point is reflected in Example 1. In that example, each predictor is one insurance company, offering an auto insurance plan based on what it predicts to be the most suitable for the driver. The company can offer discounts to attract the customer. Then the decision of the customer not only depends on the suitableness of the plan, but also how much the company is willing to pay/offer.
   I think it is important to adjust this part to a more realistic setting.

2. Predictor's decision: the paper does not discuss much about how the predictor makes the decision. On page 2, it states that "ML predictors can use any stream-based AL algorithm. For instance, a predictor can use the uncertainty-based AL rule." The predictor's decision may play a very important rule in this problem. I encourage authors to include more discussions as well as numerical experiments on how different active learning algorithm would lead to different/similar results.

The data distribution P_{X,Y}: Is the data distribution known to all predictors and how do they utilize this piece of information in their budget allocation? How do different distributions influence the final result? This distribution can also change over the time. It would help readers to get some interesting intuitions if authors can add some numerical experiments on non-stationary data distributions.


3. Numerical experiments: The paper studies effects of data purchase on quality in Section 3.1 and effects of data purchase on diversity in Section 3.2. Figure 2 shows QoE as a function of the overall quality in various levels of budget and \alpha.  It is interesting to observe that in Fashion-MNIST in Figure 2, the yellow curve decreases with overall quality while blue, green curves increase with overall quality. Could you provide more intuition of that? In Skin-nonskin in Figure 3, red curve increases with the budget. Why does that happen?

As mentioned in point 2, can you implement different AL algorithm on these datasets to verify the effectiveness and performance of other AL algorithms?

Minor Comments

- To help readers to understand easily about the problem setup etc., the paper should provide more intuitive examples.
- Probably it is better to move Section 5 Related works right after Section 1 as Section 2.



**Strengths And Weaknesses:**

Overall, I think the paper studies an interesting problem, but I have following comments that I hope authors can address in the revised manuscript.

Strengths:

1. The problem setup is interesting;
2. The paper is well-written;
3. Numerical findings are interesting.

Weakness:

1. The modeling part, especially the modeling of the predictor's decision is weak;
2. The paper does not have very strong theoretical support.

Please see my major comments below.

---

> ### Author Response · Authors · 2022-09-06
> **Response to Reviewer WCny**
>
> We appreciate the reviewer for the constructive comments and suggestions.
>
> **Problem setup:** We have included two different extensions from the submitted paper. Please see the general response.
>
> **Data distribution:** We assume that any information about data distributions is not known to predictors except user data are i.i.d. samples. In our experiments, different datasets represent different distributions, and the main patterns are consistently observed. The non-stationary data distribution is an interesting setting but beyond our scope of analysis.
>
> **Numerical experiments:** Thank you for the suggestion. We would like to clarify that QoE is not always negatively related to the overall quality. Although our main finding is the negative correlation between QoE and the overall quality, it is actually possible that QoE increases when the overall quality increases. It happens when $\alpha$ is not large enough, and this is theoretically explained with the lower bound condition $\alpha$  in Theorem 1. The intuition is that when users are not selective enough, then the QoE is mostly affected by the overall quality. However, when users are selective, then users can better find the highest quality prediction even though the overall quality available is low.
>
> **Different active learning algorithms:** In the revision, we conduct additional experiments with different active learning algorithms. Specifically, we consider confidence-based active learning algorithms, i.e., competitors decide to buy a user datum based on the confidence of their model prediction. Here the confidence is evaluated at the maximum of predicted probabilities over different classes. We observe a consistent pattern with different active learning algorithms. We have included a detailed discussion in Appendix B.3 (page 15).
>
> **Related works:** We move the ‘Related works’ section to the end of Section 1 (pages 2-3).

---

### Review · Reviewer_j99t · 2022-08-16

**Summary Of Contributions:**

The paper studies the dynamics of the process of data acquisition in an environment in which multiple agents are providing an ML service and want to improve their service thanks to new data. The paper formalizes this multi-agent setting and provides simulations to show the effect of the interaction among the agents. Finally, they also provide some theoretical results on the evolution of the system in a specific case in which two distinct sets of ML services are present in the market.

**Broader Impact Concerns:**

I think that the discussion about the broader impact is correct and does not require substantial modifications.

**Requested Changes:**

"To analyze ... predictions." This is not clear to me. From the description you provide in the following paragraph, you are selecting at random if anyone is interested in buying the data. Why not choose, even in this case, based on the performance of the classifier?

Is there empirical evidence that the user chooses the service according to Equation (1)? Otherwise, it seems like you made an arbitrary choice of this function.

Please define some symbols for the metrics in the experimental part, so they can also be used in the figures as labels.

The difference between this work and the one by Ginart et al. 2021 is stated in the related works, which are placed at the end of the paper. On the one hand, since you did not provide any technical comparison in this section, I would suggest moving it to the first part of the paper to give the reader a clear idea of the novelty of the proposed work. On the other hand, I think a comparison in terms of theoretical results with the ones of Ginart et al. 2021 would be a nice addition to the paper.

The choices of the parameters in the experimental part are not justified or motivated. This reduces the value of your findings since it may happen that in real-world situations, the behaviour might be different. I suggest justifying how you choose these parameters.

"As the evaluation time ... rounds." This concept is not clear, please rephrase.

"In other words, ... subgroup." It is not clear to me what you mean by subgroup.

It is not clear to me why you decided to move the lower-upper bounds on the QoE in the appendix. I suggest you move them to the main paper.
Moreover, I suggest providing more insights and correspondence with the empirical findings you presented. The same holds for Theorem 1. I think it would be better to refer to the specific result to compare this result and the one obtained in the previous section.

Is the experimental setting fulfilling all the assumptions needed for Theorem 1? How do you define the two sets F_1 and F_2?

**Strengths And Weaknesses:**

The paper is clear but would benefit from the use of a more detailed formalization of some quantities used in the paper, especially in the experimental part. Even if the model used is clearly described, the modeling choices are not always clear and require either an empirical motivation or reference to motivate their use.

I think this work has a valuable interest from an application point of view. The outcomes provided by the experiments are an interesting starting point for the design of algorithms reaching some kind of equilibrium under specific budget constraints.
Conversely, this work would greatly increase its interest by the scientific community if some theoretical analysis of the dynamics of the system is added. I think this lack reduces the novelty of the paper by a great margin.
Finally, the authors provided some discussion about the similarities with the MARL field. I suggest expanding such a discussion and clarifying more the difference between what is presented here and the existing MARL settings.

---

> ### Author Response · Authors · 2022-09-06
> **Response to Reviewer j99t**
>
> We thank the reviewer for the detailed comments and thought-provoking feedback.
>
> **Regarding MARL:** We have added a few more references on sublinear regret algorithms on MARL and a literature review (page 3).
>
> **Relaxing the assumption on user selection:** Please see the general response.
>
> **Justification for Equation (1):** Inspired by Ginart et al. (2021), we considered the same user selection model when no competitor shows purchase intent. Equation (1) reflects that users are more likely to choose the one with a higher quality prediction.
>
> **Comparison with Ginart et al. (2021):** Theorem 4.4 of Ginart et al. (2021) examines the effect of the number of competitors on QoE, but our theoretical result focuses on the relationship between the overall accuracy and QoE. Theorem 1 explains why QoE decreases when ML predictors can actively acquire user data through the data purchase system, supporting our main findings in experiments. We have added this note in the revision (page 11).
>
> **Lower and upper bounds on the QoE:** Thank you for the suggestion. We moved it to Section 4.1 of the revision (page 11).
>
> **Feasibility of the assumptions in Theorem 1:** Please see the general response.
>
> **Editorial issues:** Thank you for pointing out these issues. We have clarified the explanations in the revised paper.

---

### Review · Reviewer_WG6Y · 2022-08-28

**Summary Of Contributions:**

The paper studies the impacts of data purchase in ML competition; it first constructs a competition environment where a set of predictors interact with data owners over multiple rounds: at each round, a data owner reveals the unlabeled data to predictors who then decide whether to purchase the data with their budgets or not. If at least one predictor shows purchase intent, then the data owner will select one uniformly; otherwise, the data owner will receive the prediction from predictors with probability proportional to the prediction accuracy. For both cases, only the predictor that got selected receives the true label. Based on this environment, the paper empirically examines the impacts of data purchase on the predictor quality, the quality of user experiences, and diversity; and it theoretically identifies conditions under which the quality of user experience diminishes as individual predictor quality improves.


**Requested Changes:**

1. The user quality of experience (QoE) is only calculated based on the settings when there is no buyer, and it seems that the users with buyer(s) will NOT receive the predictions. More justification on this is needed (i.e., why does QoE not depend on the settings with buyers). In example 1 (auto insurance) given by the authors, if the company attracts a user by offering a discount, doesn’t the user also receive the insurance plan from this company?
2. I am wondering whether the model can be generalized to a setting where different predictors compete for a potential user by offering different prices, and the user selects the buyer offering the highest price (in contrast to the present model where the price for purchasing a data point is one for every predictor and the user selects the buyer uniformly). If so, how may the conclusions be affected; if not, what are the challenges?

**Strengths And Weaknesses:**

Strengths:
1. The paper explores an interesting research topic: the impacts of data purchase on ML competition. To the best of my knowledge, this is a research area that is less explored in the literature.
2. The paper establishes a formulation to model the competition environment, and conducts extensive empirical experiments on multiple real datasets to examine the impacts of data purchase on the predictor quality, the quality of user experiences, and diversity. Importantly, the theoretical results are consistent with the empirical findings.

Weaknesses:
1. My biggest concern is that the main conclusion of the paper (Theorem 1), i.e., the quality of user experience may decrease even as the quality of each individual predictor improves, heavily depends on the modeling assumption. Specifically, the paper implicitly assumes that the users do not receive predictions if there is a buyer who shows the purchase intent and the quality of user experiences is only calculated based on the settings with no buyer. However, this assumption may not be realistic. If we change the definitions of predictor overall quality and quality of user experience, the conclusion may no longer hold.
2. The sufficient conditions of the theoretical results are quite strong, i.e., the conclusion holds only when users are sufficiently confident in their selections ($\alpha\geq C_{\alpha}$), and when the variance of the set of predictors with the higher overall predictor quality is smaller. How about other settings? More results on other problem settings would make the paper a much stronger submission.

---

> ### Author Response · Authors · 2022-09-06
> **Response to Reviewer WG6Y**
>
> We thank the reviewer for the helpful comments and feedback.
>
> **Clarification on user selection:** We would like to clarify that the reviewer’s concern, “the paper implicitly assumes that the users do not receive predictions if there is a buyer who shows the purchase intent”, is not true. If there is more than one buyer, then the users will obtain one of the buyers' predictions by selecting a buyer uniformly at random. We explained this in the revised paper (Section 2).
>
> **Assumptions in Theorem 1:** We greatly thank the reviewer for this thought-provoking question. Please see the general response.
>
> **Justification for QoE:** Thank you for bringing this to our attention. The main reason we define QoE not to include the data purchase procedure is to capture the user’s expected satisfaction that is only driven by user selections after $T$ competition rounds. This definition is suitable for a fair comparison of two different competition environments that are evolved from different initial budgets after certain rounds of competition. We have included this note in Section 3 (page 5 of the revision).
>
> **Generalization of the competition environment:** We believe there are various ways of extending our framework including the reviewer’s point. In the revision, we generalize the proposed environment to two different settings. Please see the general response.

---

### Author Response · Authors · 2022-09-06
**Response to all reviewers**

We thank all the reviewers for their time and constructive feedback. We are glad that the reviewers appreciated that our framework establishes interesting competition environments and has practical values in applications. In the revised paper that we have uploaded, we have carefully incorporated the reviewers’ helpful suggestions. All the revisions are reflected in red. Also, we respond to each reviewer’s comments and the main additions are summarized as follows.

**Adding more realistic competition environments:** The reviewers recommended relaxing model assumptions to make proposed competition environments more realistic. In the revision, we consider the following two extensions:
- Each user datum has a different value.
  - Some user data can be noisier than others, and it results in different data values.
  - Previously, the price of user data is fixed to one.
- A user chooses one competitor based on their prediction quality even when there is more than one buyer.
  - A user prefers a buyer with a high-quality model prediction.
  - Previously, a user simply selects one uniformly at random regardless of its quality.

New experimental results show that our main findings are consistent under these relaxations—i.e., the prediction quality users experience can decrease even when the overall quality increases. We have included details in Appendix C of the revision (page 16).

**Assumptions in Theorem 1:** Reviewers WG6Y and j99t asked about the practical feasibility of the assumptions in Theorem 1. We empirically find that the assumptions are mostly achievable when comparing two competition environments with different initial budgets. Specifically, on the Insurance dataset, we compare two environments where one consists of competitors with $n_{\mathrm{b}}=0$, and the other has competitors with $n_{\mathrm{b}}=400$ when $\alpha=4$.
We denote a set of competing predictors by `\mathcal{F}_1` when $n_{\mathrm{b}}=0$ and `\mathcal{F}_2` when $n_{\mathrm{b}}=400$. In this setting, the constants are $(C_{\alpha}, C_1)=(0.03, 0.08)$ and the variances are $(\mathrm{Var}[Z_1], \mathrm{Var}[Z_2])=(0.01, 0.11)$, making the inequality assumptions in Theorem 1 feasible. We have included this note in Theorem 1 (page 11).

---

### Decision · Action_Editors · 2022-10-25

**Recommendation:** Accept with minor revision

**Comment:**

All reviewers are in agreement that this paper studies an interesting setting that models the impacts of data purchase on ML competitions and has practical values, providing both theoretical results and empirical simulations. They also agree that the revision and the newly added experiments in relaxed environments addressed some of their their initial concerns.

The reviewers would have appreciated a more detailed and deep discussion on some of the issues raised by them (e.g. the second bullet point of Reviewer WCny's comment 1).

**Audience:**

The reviewers think that the topic is interesting and relevant to the ML area, for example, in data purchase, ML practice, and Multi-agent RL.

**Claims And Evidence:**

The reviewers think that the claims provided by the authors reflect the content of the paper; the arguments presented by the authors sustain their claims properly.

---

> ### Author Response · Authors · 2022-11-02
> **Response to Action Editors**
>
> We thank the action editor for the positive recommendation and the comments. We also would like to thank all the reviewers for their constructive feedback during the review process. We are greatly glad that the three anonymous reviewers found our paper to have practical value and our revision addressed their initial concerns.
>
> We have uploaded our revised manuscript. In the revision, we add a discussion of limitations (e.g. Reviewer WCny's comments) in the Conclusion section and present future research directions of this work. In addition, we add a GitHub link that provides instructions on how to simulate the proposed competition environments. Thank you so much.